# Framework for analyzing MAE-derived immunopeptidomes from cell lines with shared HLA haplotypes

Queenie W. T. Chan[1], Teesha C. Baker[2], Chia-Wei Kuan[3], Lucy Song[1], Hongbing Yu[4], Leonard J. Foster[1]*

1 Department of Biochemistry & Molecular Biology, Michael Smith Laboratories, and Life Sciences Institute, University of British Columbia, Vancouver, BC, Canada, 2 Department of Pediatrics, BC Children's Hospital Research Institute, University of British Columbia, Vancouver, BC, Canada, 3 Agilent Technologies, Santa Clara, California, United States of America, 4 Department of Microbiology, Molecular Genetics, and Immunology, University of Kansas Medical Center, Kansas City, Kansas, United States of America

* leonard.foster@ubc.ca

## Abstract

### Background

The goal in vaccinology is to identify candidate antigens for clinical trials that will elicit an immune response for a significant portion of the target population. Unfortunately, promising data generated at the preclinical level often cannot be replicated in larger sample sizes. The goal of this project was to develop a methodology for processing MAE-generated data to identify MHC epitopes, minimize non-specific contaminants, find binding motifs, and utilize genetic connections among donors to determine which peptides were presented by specific MHC alleles.

### Results

Our approach demonstrated that mild acid elution of peptides from seven consanguineous B-lymphocyte lines accurately reflects the HLA genotypes within family members, highlighting the specificity of MAE. Additionally, the data successfully reproduced known MHC binding motifs and partially deconvoluted the originating HLA alleles of the epitopes.

### Conclusions

These findings suggest that our approach could be applied to numerous cell lines globally to evaluate a wide array of HLA haplotypes. This may help to reveal candidate vaccine antigens that induce immune protection for a wider population.

**Data availability statement:** All mass spectrometry raw data, search files, and in-house scripts have been deposited to the ProteomeXchange Consortium via the PRIDE partner repository with the dataset identifier PXD058267 (https://proteomecentral.proteom-exchange.org/cgi/GetDataset?ID=PXD058267).

**Funding:** The author(s) received no specific funding for this work.

**Competing interests:** The authors have declared that no competing interests exist.

## Introduction

Vaccines are the best preventative measure against infectious diseases, having saved millions of lives at relatively little cost. They are also the only conceivable way of completely eradicating a disease [1], benefiting the health of both current and future generations. Despite advancements such as subunit and mRNA vaccines, there are still many diseases without effective vaccines, suggesting that more systematic approaches must be taken. Currently, "reverse immunology" is the conventional method for discovering subunit vaccines: hundreds or thousands of potential antigens are synthesized and tested individually for their ability to elicit an immune response in animal models. Not only is this a time-consuming process, but there is little guarantee that the results will translate to humans. Even existing vaccines do not elicit protective immunity in all recipients, with some at efficacies of 50% or lower [2]. One major reason for this is the unparalleled genetic diversity of MHC proteins among the human population [3]. This might be overcome by testing vaccine candidates in more individuals in the early phase of testing, especially selecting participants from the vaccine's target demographic, but that would be financially unworkable. Nevertheless, this is at least, in principle, a crucial aspect of evidence-guided, rational vaccine design. A more practical approach would be to instead use many human-derived cells from this population to identify immunodominant antigens prior to clinical testing. With the rapidly improving techniques surrounding immunopeptidomics, i.e., the mass spectrometric analysis of MHC-bound peptides, this can be a real possibility.

Traditionally, antigen discovery by immunopeptidomics involves using antibodies that selectively precipitate MHC proteins from lysed cells, then eluting presented peptides for identification by mass spectrometry (MS). This method is widely used but it requires significant time, effort, and expertise, making this impractical to apply to many cell types. Immunoprecipitation also tends to select only for strongly bound peptides, which may not be the most immunogenic *in vivo* since immunodominance is also highly affected by protein abundance [4]. The procedure also requires multiple immunoprecipitation reactions with different antibodies to cover all MHCs that might be present in a system, and cell lysis prior to the immunoprecipitation means that all MHC-bound peptides are sampled instead of just the ones on the cell surface. Alternatively, the mild acid elution (MAE) technique, where peptides are stripped directly from the cell surface, is growing in popularity [5]. While this technique is relatively easier and quicker to carry out, the complicated variety of antigen-presenting proteins [6] and other proteins on the cell surface and in the extracellular milieu means the results are more complicated to interpret.

Here we present an analytical framework for MAE-based immunopeptidomics that is fast, sensitive, and specific and can be applied to any antigen-presenting cell. This can pave the way to study many cell lines from a vaccine's target population, analogous to recruiting a large number of trial participants. To illustrate this concept, we tested it on B lymphocyte cell lines from a family of seven – father, mother, and five children – discovering which antigen consensus sequences are most likely to be immunodominant for this group. Although this is a small sample size, the shared MHC alleles among

these individuals reflect the genetic variations found within an ethnic group. Using samples collected by MAE, we focused on the narrow length of MHC I-presented ligands and the tendency of MHC II to present nested (i.e., overlapping or "ragged") peptides [7]. The goal of this project was to develop a methodology for processing MAE-generated data to identify MHC epitopes while minimizing non-specific contaminants, finding binding motifs, and test how we can take advantage of the genetic connections among the donors to determine which peptides were presented by specific MHC alleles. Ultimately, this work aims to advance the rational selection of vaccine antigens at the preclinical stage. By incorporating HLA alleles from the target population in cultured cells, researchers can address the diverse HLA polymorphisms early in the process, rather than during costly clinical trials where many vaccines fail to show immune efficacy among participants.

## Materials and methods

### Cell lines

Seven immortalized B-lymphocyte cell lines from Family 243 (father GM2705, mother GM2707, five children by birth order GM2728, GM3027, GM2713, GM2709, GM2711) were purchased from the Coriell Institute for Medical Research. Partial HLA genotypes for each were available from the Coriell Institute and these were supplemented with additional typing using HLA Fusion 3.2.0.13925 (service provided by the Vancouver General Hospital Immunology Laboratory). Cell lines LBL-721 and MHC-mutant LBL-721–174 were kind gifts from Dr. Paul Sondel (Department of Human Oncology, University of Wisconsin School of Medicine and Public Health). These cells were maintained in RPMI-1640 medium with varying percentages fetal bovine serum according to manufacturer's protocol.

### Generation of stable cell line with lentivirus infection

HLA-A*02 shRNA construct (Clone ID: TRCN0000057238, GE Healthcare) and a non-mammalian-targeting control shRNA (SHC002, Sigma) were packed into lentivirus particles in 293T/17 cells (ATCC) using the lentiviral packaging mix (Sigma). GM02709 cells, one of two B lymphocyte cell lines that possess homozygous HLA-A*02 alleles, were transduced with these lentivirus particles in the presence of 8 μg/mL polybrene. Stable knockdown cells were selected with puromycin (1 μg/mL) for two weeks.

### Mild acid elution

A total of $5x10^7$ to $2x10^8$ cells were required for each biological replicate, with a minimum viability threshold of 85%. All experiments were done in triplicate. Cells were harvested for elution by centrifuging cells (Thermo Scientific, Sorvall T1 centrifuge) at 500 x g for 3 min in 50 mL conical tubes. This and all following steps were performed at 4°C to prevent sample degradation and minimize proteolytic cleavage. Note that it is impossible to differentiate between peptide degradation due to sample handling and natural antigen processing by the immunoproteasome or cathepsins [8]. Cell pellets were washed sequentially with 10 mL of room temperature 1X PBS and then twice with 10 mL with cold PBS. To remove residual phosphate salts, cells were then resuspended in 10 mL of cold saline, made with the same sodium chloride and potassium chloride concentration as 1X PBS, but without either of the cations' phosphate salts, then transferred to new conical tube. Following the same centrifugation conditions, the supernatant was once again discarded. The elution of presented antigens was achieved using a solution of 2% acetic acid in the same 1X cold saline as above. Cells were pelleted at 1000 x g for 5 minutes. The immunopeptide-supernatant was collected in a new tube then frozen at −80°C overnight, or snap frozen in liquid nitrogen then lyophilized until dry.

### Preparing peptides for MS analysis

**Desalting peptides.**  Immunopeptide samples were thawed or resuspended in 1 mL of Desalting Buffer A (2% acetonitrile in 0.1% TFA), ensuring that the sample has been acidified to below pH 2.5. Using the Stop-And-Go Extraction

(STAGE) tip desalting protocol [9] desalting columns were prepared by cutting out small cores (using a 14-gauge flat-tipped syringe needle) from Empore C18 solid phase extraction discs and inserting them into P200 pipette tips. The C18 material was made wet by passing 50 μL of methanol, using the pressure generated by applying force on the plunger of a 10 mL plastic syringe to push the liquid through. The columns were then conditioned by passing 100 μL of Desalting Buffer A. Peptide samples were applied to the column, then washed twice with 200 μL of Desalting Buffer A. Peptides were eluted with 80 μL of Desalting Buffer B (30% acetonitrile in 0.1%) into a clean 96-well plate or microtubes. Finally, the samples were dried by vacuum centrifugation before proceeding to the next step.

**Offline liquid chromatography fractionation.** Desalted and dried samples from the LBL-721 and LBL-721–174 cell lines were resuspended in 23 μL of Fractionation Buffer A (2% acetonitrile in 5 mM $NH_4HCO_2$, pH 10) and fractionated through a 36 min gradient on an Agilent ZORBAX Extend 80 Å C18 column. The separation was performed at high pH, using a gradient that ran from 0% Fractionation Buffer B (90% acetonitrile in 5 mM $NH_4HCO_2$, pH 10) to 4% B over 30 seconds, then increased to 40% at 20 min, held at 90% for 5 min, and equilibrated for the remaining 10.5 min at 0% B. Each sample was separated into 12 fractions (~120 μL per well) in a 96-well plate then desalted and dried once more.

**Sample resuspension.** Prior to MS analysis, samples were reconstituted in 20 μL of MS Buffer A (2% acetonitrile in 0.1% formic acid). A NanoDrop One (ThermoFisher Scientific, A205nm, Scopes mode) was used to spectrophotometrically quantify the amount of peptide in 1.5 μL of the sample. Peptides were diluted to 1 μg/μL for a 2 μL injection in MS analysis.

### Online liquid chromatography and mass spectrometry (LC-MS/MS)

**LC-MS/MS of peptides from LBL-721 and LBL-721–174 cell lines.** Samples were reconstituted in 20 μL of MS Buffer A (2% acetonitrile in 0.1% formic acid). Purified peptides were analyzed using a timsTOF Pro (Bruker Daltonics) mass spectrometer on-line coupled to an Easy nano LC 1000 HPLC (ThermoFisher Scientific) with a C18 column using a Captive spray nanospray ionization source (Bruker Daltonics). Samples were separated with a 45 min gradient, run from 5% MS Buffer B (90% acetonitrile in 0.1% formic acid) to 30% B over 45 min, then increased to 100% B over 2 min, held at 100% B for 13 min. The mass spectrometer was set to acquire in a data-dependent PASEF mode with fragmenting the 10 most abundant ions, including +1 ions by drawing the ion mobility zone of interest to include +1 ions (one at the time at 18 Hz rate) after each full-range scan from m/z 100 Th to m/z 1700 Th. The nano ESI source was operated at 1900 V capillary voltage, 3 L/min drying gas and 180°C drying temperature. Funnel 1 was set at 300 V, funnel 2 at 200 V, multipole RF at 200 V, deflection delta at 70 V, quadrupole ion energy at 5 eV, low mass at 200 Th, collision cell energy at 10 eV, collision RF at 1500 V, transfer time at 60 μs, and pre-pulse storage at 12 μs. PASEF was on with 10 PASEF scans for charges 0–4, Target intensity 20000 and Intensity threshold 2500.

**LC-MS/MS of peptides from Family 243 cell lines.** Samples were separated using an EASY-nLC 1000 system with a C18 column coupled to a Q-Exactive mass spectrometer (Thermo Scientific). Peptides were eluted with a gradient from 100% buffer A (0.5% acetic acid) to 40% buffer B (80% ACN, 0.5% acetic acid) over 142 min at a constant flow of 250 nL/min. The mass spectrometer operated in a data-dependent acquisition mode, with fragmentation of the five most abundant ions per scan and dynamic exclusion of 30 seconds enabled. To maximize peptide identification, fragmentation was allowed for ions of +1 charge (default settings exclude +1), since the peptides were not digested by trypsin and therefore does not always have a +2 charge in an acidic environment. MS resolution was set to 70000 with an automated gain control (AGC) target of $3\times10^6$, maximum fill time of 20 ms and a mass window of 300–2000 m/z. Higher collision dissociation (normalized collision energy 26 with 20% stepping, done in accordance to previous findings to obtain optimal spectra [10] was performed with an AGC target of $1\times10^6$, maximum fill time of 120 ms, mass resolution of 70000, and charge exclusion set to unassigned.

**Processing raw data to identify peptides.** All mass spectrometry raw data have been deposited to the ProteomeXchange Consortium via the PRIDE partner repository [11] with the dataset identifier PXD058267.

## Analysis of MAE data from HLA-A*02 knockdown cells

Raw data from HLA-A*02 knockdown and control cells were searched using MaxQuant (v 1.4.1.2) [12]. Default values were selected with these notable exceptions: Match Between Runs checked, unspecific digestion mode (i.e., no enzyme), no fixed modifications, N-terminal protein acetylation and methionine oxidation as variable modifications, reverse decoy mode, initial search using a human first search database provided by MaxQuant, default contaminant database used, peptide-spectrum match and protein false-discovery rate (FDR) of 0.1. The search was conducted against a protein database containing UniProtKB/TrEMBL human sequences (88844 sequences), cow sequences (2151 proteins, taken from bovine 32006 entries as of October 2013) that were identified in a separate search against cow sequences alone, and 11 likely viral contaminants including the Epstein-Barr virus, adenovirus, and bovine diarrhea virus.

Peptides found in at least 2 of 3 biological replicates were considered identified to produce a confident and reproducible dataset. After normalizing the peptide intensities, we calculated a knockdown/control presentation ratio for all peptides of human origin. Peptides found in the knockdown cells only were arbitrarily assigned an arbitrary but high ratio of 50, and ones found only in the negative control were given the reciprocal ratio of 0.02. The average remaining HLA-A*02 protein after knockdown was 21%, as estimated by western blot so we used this knockdown factor as a cut-off to determine whether a given peptide was or was not affected by the knockdown. Of those that were affected, 87% were less than 12 residues long, suggesting that they are MHC I peptides and consistent with the knockdown of one of the MHC I genes.

## Analysis of MAE data from LBL-721 and LBL-721–174 cell lines

Data was searched using Byonic software (Dotmatics) against a database composed of all human annotated and reviewed protein entries (Uniprot, Swiss-Prot). A database of bovine proteins (Uniprot, Swiss-Prot) was added to a standard decoy database of common contaminants. A non-specific *in silico* digestion was performed with peptide lengths from 7–25 amino acids. Parameters used included precursor mass accuracy of 70 ppm and fragment mass accuracy of 40 ppm. A common variable modification of methionine oxidation was included. A 1% FDR cut off was used at the protein level with the software automatic score cutoff applied at the peptide level. A Byonic peptide score of 2, approximately corresponding to a *p*-value of 0.01, was used to remove low confidence identifications.

## Analysis of cell lysate data from LBL-721 and LBL-721–174 cell lines and MAE data from family 243 cells

Data was searched using Fragpipe [13] version 19.1 against a database of Human Uniprot sequences including isoforms (104597 sequences, downloaded on May 16, 2024) with Bovine, contaminant and reverse decoy sequences. Default values were selected with these notable parameters: Match Between Runs checked, non-specific enzyme, no fixed modifications for MAE data or carbamidomethyl modification for cell lysate data, N-terminal protein acetylation and methionine oxidation as variable modifications. Other key parameters include peptide lengths set at 7–25 amino acids, precursor mass accuracy of 15 ppm and fragment mass accuracy of 20 ppm, and 1% FDR cut off was used at the protein level and peptide level, the acceptable standard used in mass spectrometry. For cell lysate data, peptides were considered identified if it was found in at least two out of three replicates. Data visualization and statistical analysis for cell lysis data was performed with FragPipe-Analyst [14] to generate a volcano plot.

## Organizing peptides into potential epitopes

To remove sequences with low technical confidence, all epitopes (whether derived from single or overlapping peptides) with a total spectral count of 2 or fewer were removed from further analysis. Peptides with 8 or fewer residues were also rejected. Peptides whose sequences overlapped with others were condensed into the longest possible outcome and considered as epitopes. Peptides that did not overlap with any others were also considered as epitopes. Epitopes measuring 9–11 amino acids were sorted into the MHC I group, and 12–20 mers were placed into the MHC II group.

## Multiple sequence alignment

Clustal W2 [15] was used for multiple sequence alignment (MSA), selecting the BLOSUM matrix for slow pairwise alignment and the GONNET matrix for multiple alignment, with gap open penalty of 100 and gap extension penalty of 10 for both cases.

## Amino acid positional score and positional threshold score

Using the text outputs provided by the Weblogo tool [16], we extracted the following parameters: weight (W), entropy (E), its lower limit (L) from each position in every sequence logo, the number of occurrences of amino acid X ($A_x$) and the total number of amino acids at position n ($N_n$). Then an Amino Acid Positional Score (S) and Positional Threshold Score (R) was calculated using:

$$S \; = \; (A_x/N_n)^* \; E^* \; W \tag{1}$$

$$R \; = \; E - L \tag{2}$$

Alignment positions where no residues exceed the threshold are given a null value, indicated by a dash in the output. Positions with significant residue(s) are written out in standard regular expression format, with bold font, and grayscale formatted as before with the highest Amino Acid Position Score, capped at 1.0 which is defined as 100% black, and all other values in decreasing darkness with all values below 20% displayed as 20% black. Residues which only surpass the Positional Threshold only when grouped by their physiochemical properties are displayed in the same format, but with a smaller, non-bold font. Summarized views of the consensus sequences are made by displaying only the residues with top 4 Amino Acid Positional Scores.

## Results

### MAE is capable of isolating MHC I and II epitopes

To confirm the efficacy of MAE for isolating MHC I-presented ligands, we performed a shRNA knockdown of HLA-A*02 (S1 Fig, see S1 File for original blot) on the immortalized B lymphocyte cell line GM2709 [17,18] which is homozygous for this allele. Peptides were placed in the MHC I (9–11 residues) or MHC II (12–20 residues) group. Peptides with overlapping sequences were aligned and condensed, with their total span in length treated as epitopes for further analysis. Using these simple criteria, the final FDR was effectively zero as there were no hits against the reverse sequence database after applying these multiple constraints. MSA [15] followed by sequence logo [16] analysis of epitopes that were diminished in the HLA-A*02 knockdown compared to the control and revealed a consensus sequence that was identical to that known for HLA-A*02 (Fig 1A). Furthermore, 58% of the epitopes that were directly affected by the knockdown were marked as specific binders (strong or weak) by NetMHCpan [19], with another 20% designated as coming from the other MHC I molecules (HLA-B*07:02, HLA-B*35:01, HLA-C*07:02). Assuming that the remaining 22% are cell surface contaminants, this suggests that MAE can effectively produce peptide samples that are specific to MHC I.

To test the effectiveness of MAE for MHC II-presented peptides, we performed a comparable experiment on a gamma irradiated B-cell line LBL-721–174 [20,21] compared to its WT counterpart LBL-721 [22,23]. The mutant was missing 241 MHC I epitopes (71% of total detected in the 9–11 mer range, Fig 1B), consistent with a comparable (88–97%) reduced expression of major MHC I proteins (Fig 1C, S2 Fig). As for epitopes in the 12–20 mer (MHC II) group, one would expect to find no epitopes in the mutated cells, since according to next-generation sequencing results these cells lack both alleles of HLA-DRB1, HLA-DPA1, HLA-DPB1, HLA-DQA1, and HLA-DQB1 (S3 Fig). MS data indicates that there was still about 5% expression of these MHC II proteins (Fig 1B, S2 Fig). As such, we were surprised to detect 218 epitopes that account

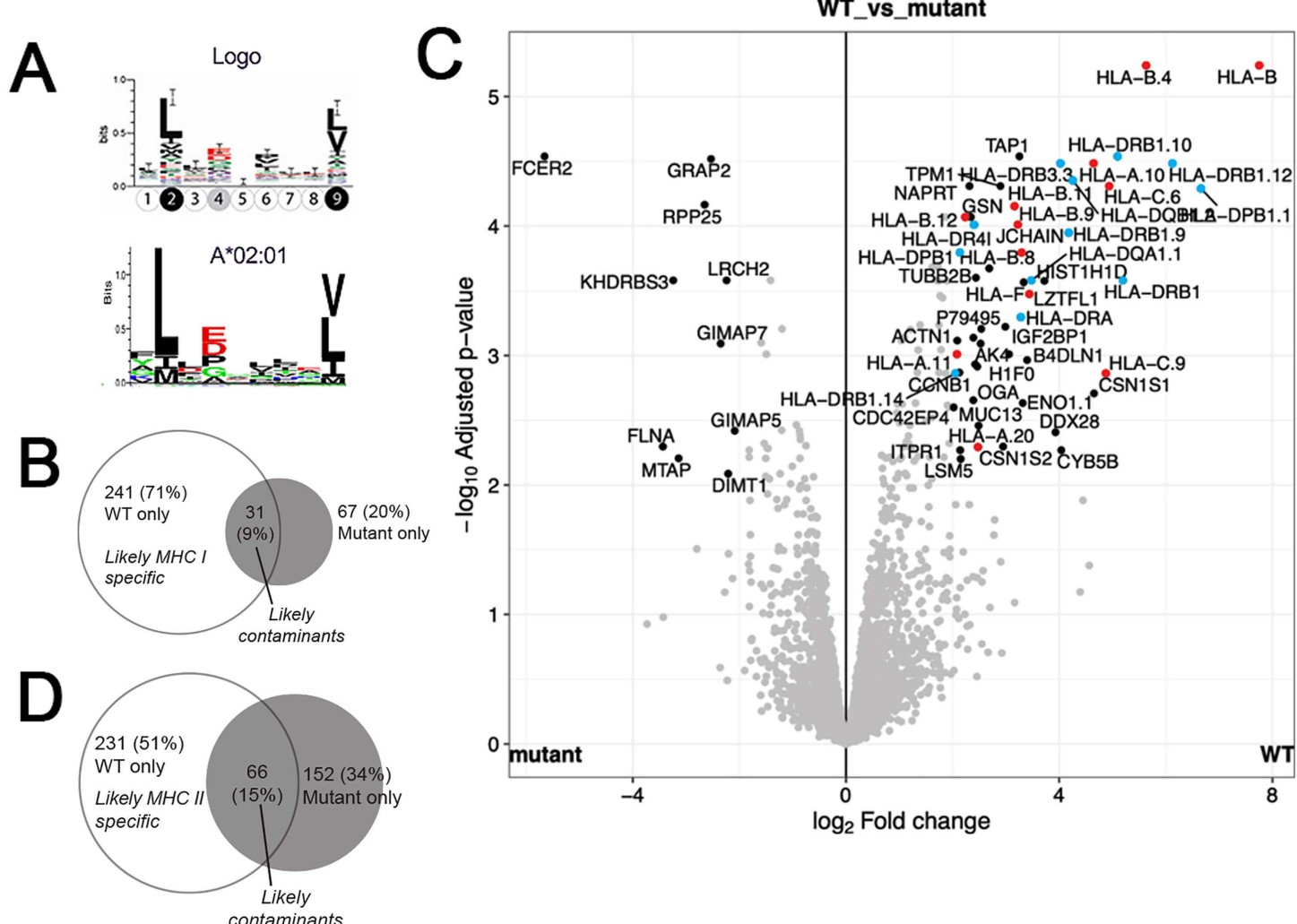

**Fig 1. Using MAE to isolate MHC I and MHC II ligands.** (A) Sequence logo (top) generated from epitopes that were found in the WT cell but not in the HLA-A*02 knockdown closely matches the HLA-A*02:01 motif from naturally presented ligands (bottom, NetMHCpan). (B) A volcano plot of the log2 fold change of proteins in a whole cell lysate of WT (LBL-721) and mutant (LBL-721-174) cell lines. Proteins associated with the MHC I (red) and MHC II (blue) antigen presentation machinery are highlighted by colored dots. Venn diagram of epitopes in the (C) 9-11 residues long MHC I and (D) 12-20 residues long category found in the WT and mutant cells.

for almost 50% of all sequences in the MHC II group (Fig 1D), implying that about half of MAE-derived MHC II peptides are non-specific contaminants. This may be because longer peptides tend to be more hydrophobic, therefore more easily adsorb to any surface [24]. This suggests that while MAE can isolate MHC II epitopes, the method's specificity must be improved by eliminating contaminating sequences.

## Benchmarking epitope sequences derived from peptide data

Applying MAE on MHC mutant or knockdown cells can differentiate between true MHC-bound peptides from non-specific contaminants. However, this advantage does not exist when we apply MAE to WT cells. In striving to filter out contaminating peptides, we removed those measuring 8 or fewer amino acids (the MHC binding groove prefers for 9-mers [7]).

To avoid "overcounting" highly similar peptides with shared sequences (produced by immunoproteasomes or antigen processing cathepsins [8], or as a result of sample handling), we coalesced them into epitopes. This avoids the illusion that a sample contains many specific ligands when in fact, many of them are just cleavage products of a larger ligand. They were classified as MHC I and II according to length as before, but MHC II epitopes must be composed of three or more overlapping peptides since this is one of their hallmark traits [25]. This approach, which was tested on seven different human-derived B-lymphocyte cell lines that have been immortalized but otherwise unmodified, is explained in detail in Methods and summarized in Fig 2A.

To evaluate whether these data filters were effective at producing an accurate set of epitopes that are relevant to the HLA alleles of a given cell line, we used NetMHCpan [19] and NetMHCIIpan [25] with default parameters to benchmark our data. Both algorithms are widely used, accommodate diverse alleles, and allow the user to input sequences of various lengths and query them against almost any MHC isotype and allele. Our goal was to produce epitope sets that contain a high percentage of true positives (TP) and minimum false positives (FP) (Fig 2B) which can be calculated as precision ($P = TP/(TP + FP)$), also known as the positive predictive value (PPV). From the seven cell lines we analyzed, the MHC I epitope set showed an average precision of 93% (min = 87%, max = 96%) while the MHC II set averaged 55% (min = 41%, max = 67%) (see S4 Fig for individual results). This prompted us to question whether there is any evidence indicating that MAE can effectively isolate MHC II-specific epitopes.

## MAE derived epitope sequences accurately reflect MHC allelic variations among related individuals

In the experiment described, we independently analyzed seven cell lines and found that some peptides were present in nearly all of them, while others appeared only in specific subsets. This was expected since these lines were derived from a single donor family, sharing HLA alleles (Fig 3A). If MAE can isolate true MHC binders, the relative amounts of their shared ligands should reflect their genetic relatedness, rather than being random. To test this, we first calculated the average spectral count (a common MS-based value that reflects peptide abundance [26] of each MAE-derived peptide from the seven cell lines, which we termed the Peptide Index ($P_i$). We then condensed the peptides into epitopes as previously described and assigned each epitope a value called the Epitope Index ($E_i$), equal to the sum of the $P_i$ values of all its constituent peptides. Epitopes were categorized into MHC I or II groups, and sequences from each group were hierarchically clustered based on their $E_i$ values using centroid linkage mode (Cluster 3.0 [27]). The clustering patterns of all nodes in the cell line dimension for both MHC I and MHC II epitopes were consistent with the HLA genotypic relatedness among the donors, showing slight differences in node distances between the two groups and the placement of the parents (Fig 3B). This strongly suggests that the method is highly specific for isolating presented peptides, with their abundance and sequence directly linked to their MHC alleles.

## MAE performed on consanguineous antigen presenting cells can be used to define HLA allele-specific consensus sequences

One key goal of immunopeptidomics is to define the consensus binding motif for individual MHC alleles. Identifying peptides presented by a specific MHC allele is impossible due to the absence of allele-specific antibodies. While MAE on a single cell line provides quick insights into immunodominant peptides, it cannot determine which MHC allele presented them. However, combining the analysis of genetically related antigen-presenting cells may correlate specific epitopes and their relative abundance with individual MHC alleles (Fig 4A). Furthermore, clustering epitopes based on their $E_i$ values should sort them into nodes, and their sequence similarity should be revealed by MSA. For MHC I, we limited alignments to a maximum length of 12 residues due to the 9–11 residue range of MHC I epitopes (example in S2 File). For MHC II, we set a maximum alignment length of 30 residues (example in S3 File), since their long lengths make it difficult to produce high quality alignments [28]. We also removed nodes with less than 9 or more than 100 MHC II epitopes since they

# A

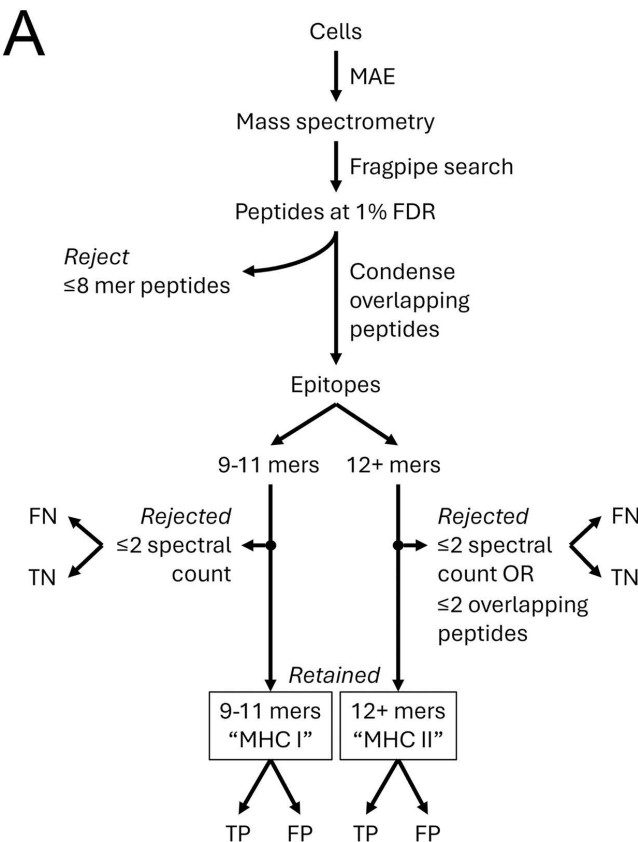

# B

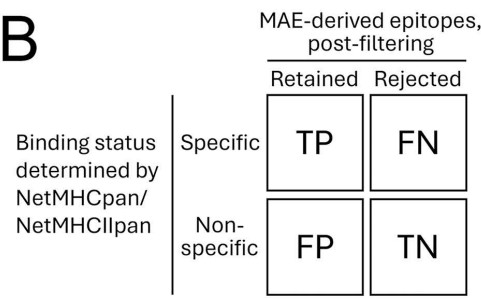

**Fig 2. Benchmarking epitope sequences derived from peptide data.** (A) A flowchart describing how MAE-derived peptide data are processed, resulting in sequences that are either rejected or retained for further analysis. (B) These sequences are then benchmarked by using NetMHCpan (for epitopes of 9-11 amino acids in length) or NetMHCIIpan (for epitopes of 12+amino acids in length) that tag entries either as a specific binder (weak or strong) or a non-binder against any of the queried MHC alleles, thereby allowing each MAE-derived sequence to be categorized as a true positive (TP), false negative (FN), false positive (FP), or true negative (TN).

also resulted in poor alignment, ultimately retaining 750 MHC I and 150 MHC II high quality nodes ([Fig 4B]). On closer inspection, sequences in some nodes displayed $E_i$ patterns that were present in some family members but absent in others, which are traceable to their HLA genes. This allowed us to identify or at least narrow down the HLA allele responsible for presenting specific ligands by comparing the experimentally derived $E_i$ profiles to theoretical profiles for MHC I ([Fig 5A]) and MHC II ([Fig 5B]).

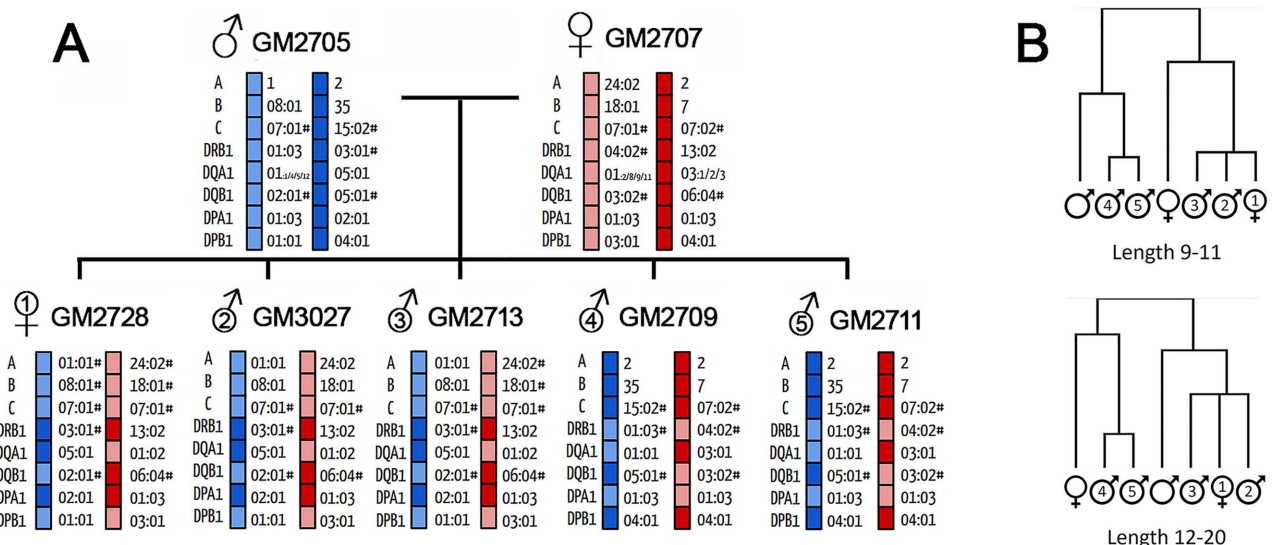

**Fig 3. HLA genetic relatedness of B-lymphocytes from a seven-member donor family.** (A) HLA alleles of each family member are shown. (B) Highly similar clustergrams were generated from $E_i$ values for both epitopes in the 9-11 mer group (top, MHC I) and 12-20 mer group (bottom, MHC II) that mirror the HLA genetic relatedness of the family members. Non-numbered individuals refer to the parents and numbered individuals are the children based on birth order.

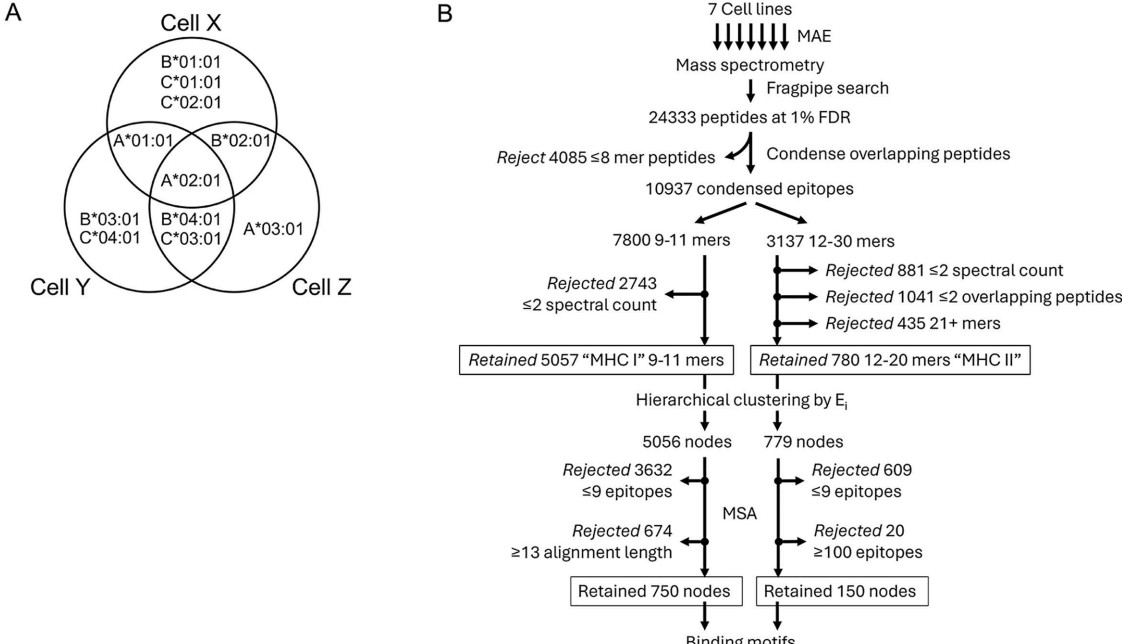

**Fig 4. Strategy for analyzing consanguineous cell lines.** (A) Analyzing the $E_i$ profiles of epitopes presented by theoretical cell lines X, Y, and Z that have partial overlap of HLA-A, HLA-B, and HLA-C genotypes can be used to trace the origin of a given epitope back to its originating allele. (B) A flow chart of how MAE-derived peptides from related cell lines can be processed together through a series of steps (condensing peptides into epitopes, hierarchical clustering, and multiple sequence alignment (MSA)) to predict MHC binding motifs.

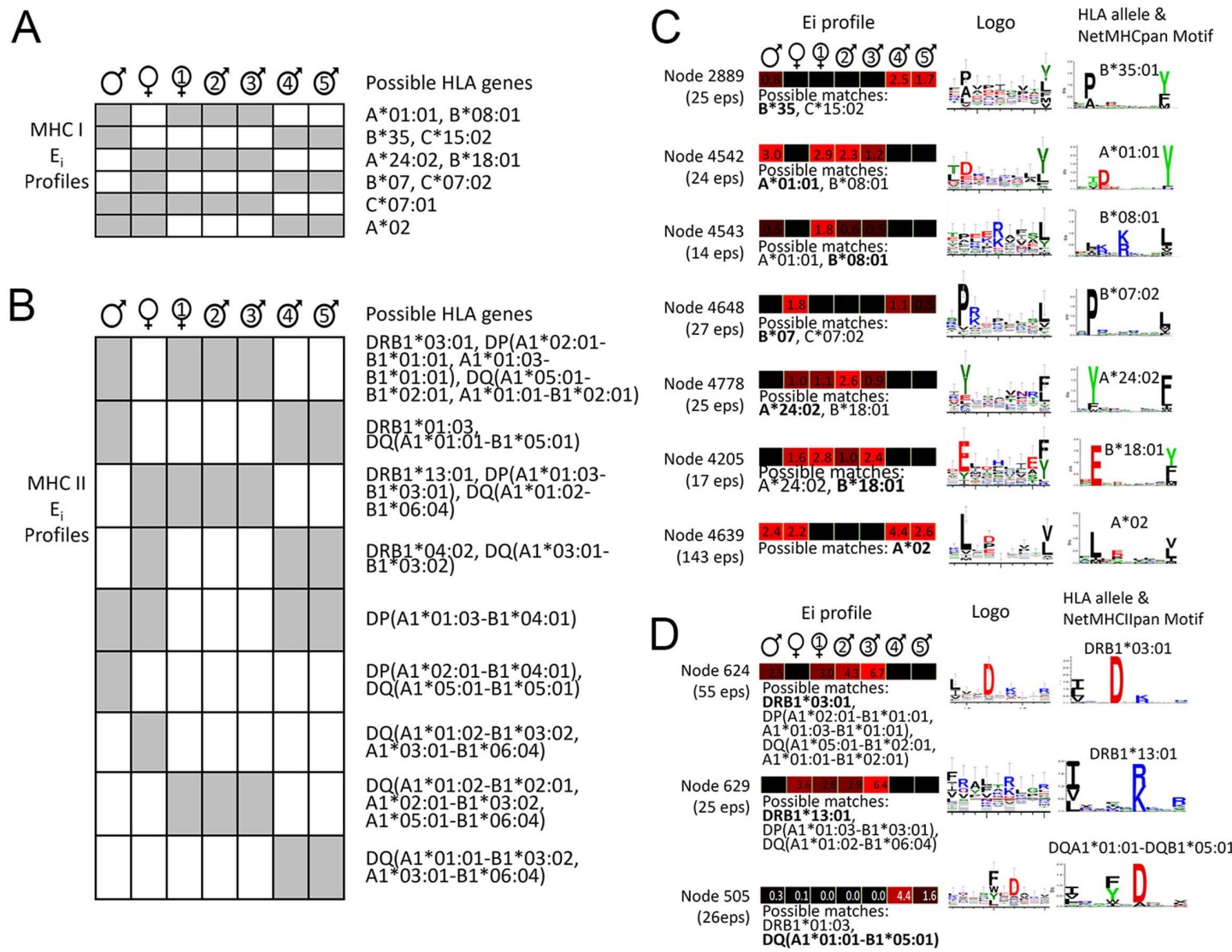

**Fig 5. Deriving HLA allele-specific consensus sequences from MAE-derived peptide data.** Theoretical Ei profiles for (A) MHC I and (B) MHC II genes for all seven members of the donor family, with non-numbered individuals referring to the parents and numbered individuals are the children based on birth order. Grey cells indicate $E_i > 0$ while white cells refer to $E_i = 0$. Positions 1-9 are predicted to be the region that fits into the MHC binding grove, with typical anchor positions highlighted (shaded circles). Shown are examples of (C) MHC I and (D) MHC II nodes that contain aligned epitopes, visualized as sequence logos, with similar $E_i$ profile family members. Epitopes in these nodes were aligned and visualized as sequence logos. Corresponding known HLA motifs from NetMHCpan and NetMHCIIpan derived from a database of naturally bound ligands [29] are shown as evidence of this technique's ability to reproduce them. Positions 1-9 are predicted to be the region that fits into the MHC binding grove, with typical anchor positions highlighted (shaded circles).

Aligned epitopes within these high-quality nodes were visualized by sequence logo plot [16], and many of them, in particular the MHC I epitopes, showed consensus sequences conforming to the major anchor sites at positions 2 and 9 of the binding core of MHC I-bound peptides, sometimes with additional auxiliary anchors. Our relatively small dataset was able to accurately reproduce every HLA-A and HLA-B consensus sequence that could be possible from these seven individuals (Fig 5C). Nodes that resulted in the weakest consensus sequences tended to contain epitopes that were found across

MAE isolates from most or all seven donors in the family (not shown), suggesting that they are contaminants. The MHC II results were more difficult to interpret because proteins such as HLA-DP and HLA-DQ are each composed of two polymorphic genes whose products can each form two *cis-* and *trans-* heterodimers. This complexity leads to weak consensus sequences, and MHC II-presented peptides are generally less studied than MHC I proteins, resulting in less reliable data for comparison. Within our data, we were able to find some MHC II nodes with sequences that form a rough consensus at positions 1, 4, 6, and 9 in the core binding region (Fig 5D). The overall lack of clear consensus sequences corroborates what is known about the more variable nature of MHC II-bound peptides.

### Visualizing consensus sequences

With the sheer numbers of nodes in this dataset, it is important to be able to find consensus sequences without needing to visually inspect each resultant sequence logo. Hence, we relied on the sequence logo algorithm to apply compositional bias compensation and Bayesian statistics at the 95% confidence level to discover consensus motifs. This allows logos (**Fig 6**, **item I**) to be summarized various one-line text outputs (**Fig 6**, **items II, III, and IV**) that can be easily viewed *en masse* or manipulated by simple scripts. Only significant amino acids at their relevant positions are shown, and non-significant positions are replaced by dashes. This effectively condenses each sequence logo into one visually intuitive line of text per node. Collapsing consensus sequences further by grouping chemically similar amino acids improves sensitivity; this is especially important for MHC II consensus sites, which are known to be more varied than MHC I.

### Evaluating the impact of epitope clustering and MSA on MHC I and II ligand precision

We wanted to test whether combining analyses of related cell lines, along with hierarchical clustering and subsequent MSA analysis could increase precision for both MHC I and MHC II ligands. These additional steps served as extra data filtering measures, removing poorly aligned epitopes that likely did not originate from the binding groove of an MHC molecule. To investigate this, we compiled epitopes from the nodes that were retained after filtering and used NetMHCpan and NetMHCIIpan to benchmark them as before (detailed results in S4 File). For MHC I epitopes, precision generally exceeded 90% for well-deconvoluted sequences, sometimes even surpassing the precision achieved when analyzing cell lines individually Interestingly, queries involving the less-studied HLA-C showed poorer performance. For instance, 38 epitopes traced back to HLA-C*07:01 achieved only 42% precision. This issue was even more pronounced for MHC II epitopes traced solely to HLA-DQ and/or HLA-DP alleles, which consistently yielded precision of 20% or less. Conversely, epitopes deconvoluted to five or fewer alleles and searched against at least one of the well-studied HLA-DRB1 molecules demonstrated precision values ranging from 42% to 87%, with an average of 60%. This represents some improvement compared to the average precision of 55% from cells analyzed independently, without hierarchical clustering and MSA. These findings suggest that the additional data processing steps have positively contributed to identifying MHC II-specific ligands.

### Discussion

Here we have shown that MAE used on antigen-presenting cells with connected genetic lineage, followed by a series of data processing steps that involve clustering epitopes by their abundance followed by sequence alignment, can help define allele-specific consensus sequences, particularly for MHC I proteins. Furthermore, this approach enables the deconvolution of these MAE-isolated peptides to their respective MHC molecules without the use of mono-allelic or knock-downs cells with reduced complexity in the HLA genes they express.

Historically, the majority of immunopeptidomic data has been generated with immunoenrichment methods. The use of MAE has only slowly increased in the past decade as there have long been doubts about its lack of specificity [5,30] regarding the coelution of non-MHC-bound peptides. One paper [30] suggested that only 40% of an MAE isolate are MHC I peptides while the rest are contaminants, yet our data from the LBL-721–174 mutant cell line indicates that the number is closer to 70%. It has also been said that the acidic environment of MAE is not enough to elute MHC II peptides [5,31].

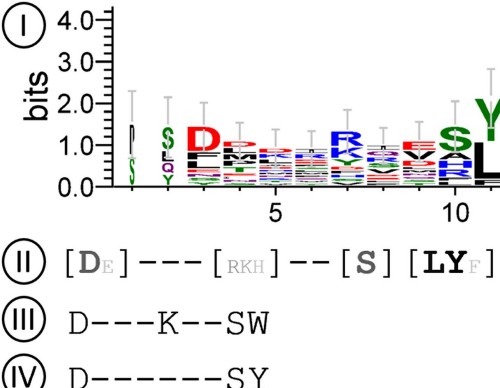

**Fig 6. Different ways to display a sequence logo.** One example node with epitopes (**I**) displayed as a sequence logo, as a single line output with (**II**) relative Amino Acid Positional Score (S) in grayscale (white=0%, black=100% or if S>1) for amino acids that exceed threshold R (bold), or amino acids that pass threshold only when "grouped" by their physiochemical properties (not bold). (**III**) Shows only the amino acids with the top 4 highest S and also pass threshold, and (**IV**) similar to (III) but also displays non-significant positions contained within the consensus are represented by dashed lines.

However, if this was true, MHC II epitope clustering by $E_i$ values would not have been able to replicate the HLA genetic relatedness between a family of cells lines. This provides some support for the use of MAE for isolating immunologically relevant ligands from antigen-presenting cells, even though its specificity towards MHC II epitopes is clearly inferior to MHC I epitopes. This problem is not a new one [32] – identifying a 9-mer core region within the longer peptide lengths of MHC II epitopes by comparing against known ligands is prone to alignment errors that lowers the accuracy of prediction algorithms [28]. This also highlights a potential downside of relying on tools such as NetMHCpan and NetMHCIIpan for benchmarking epitopes' binding specificity, since their accuracy is highly dependent on the quality of the training data taken from epitope databases [33]. When we tested these tools using hundreds of randomly generated soluble peptides [34] of 9–11 (S5 File) or 12–20 residues (S6 File) against the HLA haplotypes of our seven cell lines, NetMHCpan falsely tagged an average of 13% of these negative control sequences as a specific ligand of at least one of the queried MHC I alleles, while NetMHCIIpan erroneously labeled 15% as specific binders for at least one relevant searched allele or allele pair. Clearly, using informatics tools to predict ligands or benchmark experimental data comes with an innate possibility for error. One can reasonably argue that the immunoprecipitation method, which yields a purer sample of MHC ligands [35], is a better way to validate MAE-derived epitopes. Nevertheless, the ultimate validation is the use of biological assays that test for CD8 T cell activation by measuring cell proliferation or cytokine release. These tests are critical to linking any immunopeptidomics-based ligand discovery methodology with vaccine design. Epitopes that we identified here in this manuscript have not been validated by functional tests for their immunogenicity, and as such, our approach can only be regarded as a ligand and binding motif prediction tool at this time.

One of the most surprising aspects of the work presented here is that our relatively small dataset was able to recapitulate virtually all the known binding motifs for the relevant MHC I alleles that have been built up from a decade or more of dozens of labs collecting immunopeptidomic data [33]. Results were less clear for MHC II, yet the lack of clear consensus is not necessarily a failing of the methodology but reflects the ability of the MHC IIs' binding groove to accept a variety of secondary structures that contributes to the immune system's sensitivity for a range of foreign antigens. This propensity for promiscuous binding implies that there may be no real way to limit contamination in MHC II MAE isolates. However, since the quality of tools to isolate and analyze MHC II presented peptides continues to lag far behind those for MHC I, any attempt in improvement should be taken as moving towards right direction. Furthermore, the work here is only demonstrated through a single family of seven donor cell lines, and their genetic bias with highly overlapping HLA genotypes

among its members rather limits our methodology's ability to deconvolute MAE-derived epitopes back to their originating MHC. A better approach would be to use cell lines that share some alleles but fewer than that among parents and children. Investigating cell lines sampled from a defined ethnic group should fit the purpose, since they tend to express some alleles in common while maintaining a degree of genetic diversity. To illustrate how this can be highly effective, using our approach on merely 10 more cell lines to supplement the existing seven-donor family data will enable the epitopes to theoretically be traced back to an additional 13 MHC II alleles, up from just one from investigating exclusively the seven consanguineous cells (Fig 7, details in S7 File).

Expanding our research to a much larger population by sourcing hundreds of cells from the 1000 Genomes Project [36] is not merely an aspiration but is already underway at our laboratory. MAE has a significant speed advantage over traditional MHC immunoenrichment that it becomes vital to such a large undertaking. While antibody-based ligand isolates will always be higher in purity [39], another benefit of MAE is the elution of ligands in their native environment (i.e., embedded on the plasma membrane), and that the isolated peptides reflect their natural abundance on the cell surface. HLA proteins are not expressed in equal amounts (as shown in S2 Table and elsewhere [37]). So, while it is unfortunate that we did not detect any consensus sequences associated with HLA-C (expressed in low levels with roles in pregnancy [38]) and HLA-DP (poorly defined and implicated in some autoimmune diseases [39]), MAE-derived data perhaps make better representations of immunodominant and highly abundant epitopes that are consequential for vaccine development. Admittedly, non-specific peptide contamination will always be a major disadvantage in MAE [30,35,40]; especially for MHC II ligands, stringent filtering with epitope clustering have made only small improvements in this regard.

This study introduces a novel data analysis strategy for MAE-derived immunopeptidomes, demonstrating some ability to differentiate true MHC-bound peptides from contaminants, particularly for MHC I ligands. We were able to successfully recapitulate many documented HLA binding consensus sequences, so this approach can potentially refine existing ones or even predict new motifs. By applying this methodology to individuals with shared genotypes, along with epitope clustering and sequence alignment, we were able to partially deconvolute epitope origins without relying on HLA knockdowns. Taken together, insights can be gained by applying this approach to hundreds of cell lines from the global population, facilitated by the method's speed and ease. Ultimately, this work paves the way for developing more immunogenically inclusive vaccines at the preclinical stage, prior to involving human subjects in costly clinical trials.

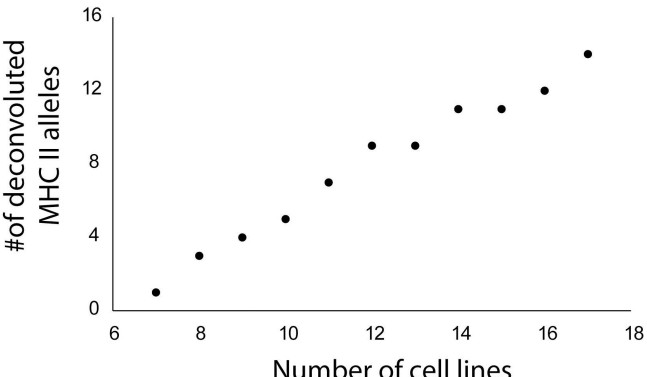

**Fig 7. Effects of adding more cell lines to the epitope clustering approach.** This graph demonstrates that for each cell line (available at the International Histocompatibility Working Group) added to the epitope clustering approach, it increases the possibility to trace a ligand back to its origin MHC by approximately one more allele.

## Supporting information

**S1 Fig. Western blot for protein HLA-A2.** shRNA knockdown of HLA-A2 in the cell line GM2709 was performed and compared against a control knockdown. Blot against calnexin served as a loading control.
(TIF)

**S1 File. Original blot of S1 Fig.**
(PDF)

**S2 Fig. Log2 fold change values of the six HLA alleles in a whole cell lysate of the WT and MHC-mutant cell lines.**
(TIF)

**S3 Fig. Haplotype data for the WT (LBL-721) and MHC-mutant (LBL-721–174) cell lines.** Shaded alleles indicate the absence of detection of that allele in the mutant cells by NGS (N/D = not detected).
(TIF)

**S4 Fig. Benchmarking MHC epitopes from individual cell analyses with NetMHCpan or NetMHCIIpan.**
(TIF)

**S2 File. Example of multiple sequence alignment of an epitope from the MHC I and aligned with no gap openings and a maximum allowable length of 12 residues.**
(TXT)

**S3 File. Example of multiple sequence alignment of an epitope from the MHC II group and aligned with no gap openings to a maximum allowable length of 30 residues.**
(TXT)

**S5 Fig. Benchmarking MHC epitopes from combined cell analysis with NetMHCpan or NetMHCIIpan.**
(TIF)

**S5 File. List of randomly generated 9–11 amino acid peptides used for benchmarking NetMHCpan.**
(TXT)

**S6 File. List of randomly generated 12–20 amino acid peptides used for benchmarking NetMHCIIpan.**
(TXT)

**S6 Fig. Cell lines added to the epitope clustering approach, in the order shown in this table, improves allele deconvolution as shown in Fig 7.**
(TIF)

## Author contributions

**Conceptualization:** Queenie W. T. Chan, Leonard Foster.

**Data curation:** Queenie W. T. Chan.

**Formal analysis:** Queenie W. T. Chan, Teesha C. Baker, Chia-Wei Kuan.

**Investigation:** Queenie W. T. Chan, Teesha C. Baker, Chia-Wei Kuan, Lucy Song, Hongbing Yu.

**Methodology:** Queenie W. T. Chan.

**Software:** Queenie W. T. Chan.

**Supervision:** Leonard Foster.

**Visualization:** Queenie W. T. Chan, Teesha C. Baker.

**Writing – original draft:** Queenie W. T. Chan.

**Writing – review & editing:** Queenie W. T. Chan, Leonard Foster.

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
