## [Decision Letter · Decision Letter 0]

15 Apr 2025

PONE-D-25-08174Towards a strategic approach to vaccine development in defined populations through mild acid elution-based immunopeptidomicsPLOS ONE

Dear Dr. Foster,

Thank you for submitting your manuscript to PLOS ONE. After careful consideration, we feel that it has merit but does not fully meet PLOS ONE’s publication criteria as it currently stands. Therefore, we invite you to submit a revised version of the manuscript that addresses the points raised during the review process.

We look forward to receiving your revised manuscript.

Kind regards,

Shahina Akter, Ph.D.

Academic Editor

PLOS ONE

Journal Requirements:

Additional Editor Comments:

Comment to the authors

The manuscript presents an innovative and valuable approach to population-specific vaccine development using mild acid elution-based immunopeptidomics. The study is well-conceived, and the results are promising, particularly regarding MHC I peptide identification. However, the manuscript would benefit from clearer articulation of objectives, justification of key methodological parameters, and improved statistical validation. The discussion could be strengthened by addressing the limitations of MAE, especially for MHC II ligands, and by comparing the method to existing approaches. Structural improvements are also needed, such as reducing overlap between results and methods sections. With these revisions, the manuscript would be significantly enhanced in clarity, rigor, and impact.

Reviewers' comments:

Reviewer's Responses to Questions

**Comments to the Author**

1. Is the manuscript technically sound, and do the data support the conclusions?

Reviewer #1: Yes

Reviewer #2: Yes

Reviewer #3: Yes

2. Has the statistical analysis been performed appropriately and rigorously? 

Reviewer #1: No

Reviewer #2: Yes

Reviewer #3: Yes

3. Have the authors made all data underlying the findings in their manuscript fully available?

Reviewer #1: Yes

Reviewer #2: Yes

Reviewer #3: Yes

4. Is the manuscript presented in an intelligible fashion and written in standard English?

Reviewer #1: Yes

Reviewer #2: No

Reviewer #3: Yes

5. Review Comments to the Author

Reviewer #1: The manuscript titled "Towards a strategic approach to vaccine development in defined populations through mild acid elution-based immunopeptidomics" is nicely presented and would be valuable addition to the literature. I congratulate the authors for conducting such nice study. However, I have a few comments, if addressed, would add up more interest for the reader.

If the authors would like, the title is informative but slightly long. It could be made more concise while retaining clarity. For example, "Mild Acid Elution-Based Immunopeptidomics for Population-Specific Vaccine Development" might be a more direct option. The abstract provides a strong summary of the study's objectives and importance. However, the methodology could be described more concisely, and key results regarding the reproducibility and validation of the method should be highlighted more explicitly.

The methodology section provides a detailed protocol, but could you please explain the justification for key parameters such as peptide identification thresholds and FDR values? Without this, it is difficult to assess the rigor of data filtering. Given that the sample selection is based on a single family, could you address the potential for genetic bias and discuss how this might affect the broader applicability of the method? Additionally, how do you control for the potential contamination of non-MHC peptides, especially in the case of MHC II ligands? While the LC-MS workflow is well described, could you provide more context on how instrument settings were optimized for peptide recovery, particularly regarding the injection volumes for different peptide concentrations? To improve reproducibility, could you include a clearer justification for analytical choices, incorporate statistical validation, and explicitly acknowledge the limitations of MAE in detecting MHC II peptides?

The results section presents strong evidence for the specificity of MAE in isolating MHC I peptides, particularly through HLA-A*02 knockdown validation, but could you clarify why the specificity for MHC II peptides appears lower? Given that a significant proportion of identified peptides may not be true MHC ligands, what steps were taken to verify their relevance? The introduction of Peptide Index (Pi) and Epitope Index (Ei) is an interesting addition, but could you provide statistical support for these indices, such as variance measures or confidence intervals? The claim that the epitope clustering approach can be applied to large population studies is intriguing, but could you explain how this conclusion is drawn from a dataset of only seven related individuals? Additionally, how do you account for the potential impact of proteolytic cleavage and peptide stability on the observed results, as these factors could introduce confounding variables? To strengthen the results, could you incorporate statistical tests for peptide abundance comparisons, include specificity controls for MHC II peptides, and temper claims about the scalability of the approach?

The discussion effectively highlights the potential applications of MAE-based immunopeptidomics for vaccine design, but could you reconsider the generalizability of the findings? While predicting allele-specific binding motifs is an important advancement, could you clarify whether these epitopes have been validated for their immunogenicity? Without functional validation, how can their relevance for vaccine development be confirmed? Additionally, could you provide a more critical evaluation of MAE’s limitations, including potential contamination, scalability challenges, and the method’s reduced specificity for MHC II ligands? Given that benchmarking against immunoprecipitation-based approaches is missing, could you explain how MAE’s performance compares to existing methods? Would it be possible to include a discussion on how this technique could be externally validated in more genetically diverse populations? A more balanced discussion should address these points to provide a clearer perspective on the method’s strengths and limitations.

Reviewer #2: The article is informative, but the presentation of results appears rushed, lacking depth and clarity. The language needs refinement for accuracy, coherence, and readability, with better word choices and sentence structures. The organization and formatting do not align well with journal standards, requiring improvements in text sequence, size, and adherence to guidelines. Figures should be of higher resolution with clearer interpretations. Substantial revisions are necessary to enhance clarity, structure, and compliance with journal requirements. The authors should address these issues before resubmitting the manuscript.

Reviewer #3: This is a well written manuscript and quite informative.

Howevver the authous should note the following.

1. It would be nice to structure the abstract.

2. The objective/aim of the study was not stated both in the abstract and the main article. In the current form , these are implied and leaves the reader in suspesnse.

3. Regarding the data analysis The statement given in the first sentence of the first pagraph ie`` on line web application`` should be referenced.

4.Regarding the results The first pagraph is talking about methods and could fit better in the methods section ,Although the results were presented well,in most sections there was inclusion of methods, sometimes discusion in the results. It would be nice to minise them so that the reusts can clearly stand out. most of the methodology statements should be included in the methodology/data analysis and the discussion statements in the duscussion section..

5.The authours discussed their results well. However they did not make deefinate conclusions from their results and nor did they make adefinate recommendations.

6. PLOS authors have the option to publish the peer review history of their article (what does this mean? ). If published, this will include your full peer review and any attached files.

**Do you want your identity to be public for this peer review?** For information about this choice, including consent withdrawal, please see our Privacy Policy .

Reviewer #1: No

Reviewer #2: **Yes: ** sarfraz ahmed

Reviewer #3: No

---

## [Author Response · Author response to Decision Letter 1]

29 May 2025

"Response to Reviewers" is located at the end of this document.

---

## [Editor Report · Decision Letter 1]

7 Sep 2025

Framework for analyzing MAE-derived immunopeptidomes from cell lines with shared HLA haplotypes

PONE-D-25-08174R1

Dear Dr. Leonard Foster,

We’re pleased to inform you that your manuscript has been judged scientifically suitable for publication and will be formally accepted for publication once it meets all outstanding technical requirements.

Kind regards,

Shahina Akter, Ph.D.

Academic Editor

PLOS ONE

---

## [Editor Report · Acceptance letter]

PONE-D-25-08174R1

PLOS ONE

Dear Dr. Foster,

I'm pleased to inform you that your manuscript has been deemed suitable for publication in PLOS ONE. Congratulations! Your manuscript is now being handed over to our production team.

Kind regards,

on behalf of

Dr. Shahina Akter

Academic Editor

PLOS ONE